# Concussion: Beyond the Cascade

**DOI:** 10.3390/cells12172128

**Published:** 2023-08-22

**Authors:** Kiel D. Neumann, Donna K. Broshek, Benjamin T. Newman, T. Jason Druzgal, Bijoy K. Kundu, Jacob E. Resch

**Affiliations:** 1Department of Diagnostic Imaging, St. Jude Children’s Research Hospital, Memphis, TN 38105, USA; kiel.neumann@stjude.org; 2Department of Psychiatry and Neurobehavioral Sciences, University of Virginia, Charlottesville, VA 22903, USA; dkb6v@virginia.edu; 3Department of Radiology and Medical Imaging, University of Virginia, Charlottesville, VA 22903, USA; btn6sb@virginia.edu (B.T.N.); tjd4m@virginia.edu (T.J.D.); bkk5a@virginia.edu (B.K.K.); 4Department of Kinesiology, University of Virginia, Charlottesville, VA 22903, USA

**Keywords:** TSPO, microglial activation, positron emission tomography, PET, concussion, neuroinflammation

## Abstract

Sport concussion affects millions of athletes each year at all levels of sport. Increasing evidence demonstrates clinical and physiological recovery are becoming more divergent definitions, as evidenced by several studies examining blood-based biomarkers of inflammation and imaging studies of the central nervous system (CNS). Recent studies have shown elevated microglial activation in the CNS in active and retired American football players, as well as in active collegiate athletes who were diagnosed with a concussion and returned to sport. These data are supportive of discordance in clinical symptomology and the inflammatory response in the CNS upon symptom resolution. In this review, we will summarize recent advances in the understanding of the inflammatory response associated with sport concussion and broader mild traumatic brain injury, as well as provide an outlook for important research questions to better align clinical and physiological recovery.

## 1. Introduction

Over the last 30 years, sport-related concussion (SRC) has emerged as a significant concern among the medical and sports communities. Although the majority (57%) of athletes diagnosed with an SRC return-to-sport (RTS) within two weeks of their injury [1], clinical measures of persisting cognitive (e.g., increased risk of mild cognitive impairment and memory deficits) [2] and motor deficits (e.g., postural control deficits) [3,4], heightened mood states (e.g., depression) [5,6], as well as other concussion symptoms (e.g., dizziness, excessive sympathetic nervous system activation, headache, and insomnia) [7] have been reported in approximately 15–20% of patients [8]. Increased awareness of persisting symptoms following a diagnosed SRC has led to foundational research on the underlying physiology of the injury to ensure athletes make a safe return-to-sport and to assuage the risk of negative health consequences.

Since 1997, a multidimensional approach has been recommended to diagnose and manage athletes suspected of having an SRC [1,9]. The recommended assessment protocol consists of clinic-based measures of balance, neurocognitive function, and concussion-related symptomology, which has been demonstrated to have a sensitivity of up to 100% within 24 h of a diagnosed SRC in collegiate athletes [10,11]. More recently, the aforementioned protocol has been expanded to include clinical measures of vision [12]. In conjunction with a clinical examination, the recommended multidimensional assessment of SRC assists with determining clinical recovery. Despite the clinical utility of this multifaceted approach, a growing body of evidence suggests that physiological recovery ensues beyond clinical recovery and return-to-sport from SRC [13,14].

Advancements in neuroimaging techniques have brought into question whether clinical recovery aligns with physiological recovery following a diagnosed SRC (Figure 1). Wang and colleagues investigated high school and collegiate football players using the Sport Concussion Assessment Tool (SCAT3), which is a multidimensional sideline assessment of SRC. The SCAT3 includes a brief neurocognitive assessment, the Standardized Assessment of Concussion (SAC), the Balance Error Scoring System (BESS), which is a time and cost-effective measure of balance, and a symptom scale [15]. The Automated Neuropsychological Assessment Metrics (ANAM) was used as a computerized neurocognitive assessment [14]. Each assessment was administered prior to injury, within 24 h, and eight days following a diagnosis of SRC. In addition to these clinical measures, participants also completed a magnetic resonance imaging (MRI) Arterial Spin Labeling protocol to measure cerebral blood flow perfusion. The authors reported significantly lower cerebral blood flow perfusion 24 h following a diagnosed SRC when compared to a control group. More concerning, the authors reported several more regions of interest that were observed to have decreased cerebral blood flow eight days following injury when the participants returned to pre-injury levels of function on the ANAM, BESS, SAC, and SCAT3 which is consistent with clinical recovery [14]. Similarly, Churchill and colleagues compared the findings of 24 collegiate athletes to a control group using the SCAT3 as well as advanced neuroimaging (global functional connectivity, arterial spin labeling, fractional anisotropy, and mean diffusivity) prior to a diagnosed SRC and at serial time points up to one year following injury. Despite each athlete returning to pre-injury levels on the third iteration of the Sport Concussion Assessment Tool 3, a multidimensional sideline assessment, and making an RTS, significantly decreased cerebral blood flow and white matter diffusivity were observed one year following injury [13].

The underlying mechanism of the “metabolic crisis” associated with SRC has been refined to include a neuroinflammatory response [22]. Following an SRC, a neurometabolic cascade characterized by an efflux of intracellular potassium and an influx of extracellular sodium and calcium occurs as a result of glutamate binding to the N-methyl D-aspartate receptor on the cellular membrane. This ionic imbalance requires adenosine triphosphate (ATP) to restore cellular homeostasis; however, cerebral blood supply has been demonstrated to be reduced by up to 50% in animal models, which limits aerobic metabolism [22,23]. Cerebral perfusion has also been observed to be decreased in human models using fractional anisotropy [14]. The well-accepted metabolic crisis associated with SRC has been demonstrated to resolve in animal models in approximately 7 to 10 days [22,23] which aligns with the clinical recovery of high school and collegiate athletes following injury [8,24]. The role of the neuroinflammatory response following SRC has yet to be elucidated. More specifically, the role of microglia following an SRC has been brought into question largely based on animal models. However, more recent human-based studies have investigated systemic inflammation via biofluid markers [19,20,25,26] and central inflammation using positron emission tomography (PET) and single-photon emission computerized tomography (SPECT) [27,28,29]. Fluid biomarkers, including cytokines (e.g., interleukin-6 [30,31], interleukin-1 receptor antagonist [20,32], and tumor necrotic factor—α [32]) and chemokines (monocyte chemoattractant proteins 1 and 4 [33,34]) following a diagnosed SRC suggest inflammation may be short-lived in alignment with standard-of-care clinical measures, such as a symptom inventory, which accounts for the quantity, severity, and duration of concussion-related symptoms [11,35], the SAC and BESS [25,31]. However, a discordance exists between these findings and those associated with central neuroinflammation. This emerging body of research brings into question the biological and clinical implications of neuroinflammation following a diagnosed SRC.

Our understanding of the neurological sequelae of SRC continues to evolve. As innovative biomarker techniques are used to investigate neuroinflammation secondary to injury in human SRC models, it is important for researchers and clinicians alike to appreciate what is known and unknown about the injury. A rigorous and critical analysis of the advantages and disadvantages of current methodologies that are being used to explore this exciting area of research is warranted. The purpose of our review is to address what is known about the underlying physiology associated with SRC, the advantages and disadvantages associated with the methodology used to investigate neuroinflammation in animal and human models and to discuss the biological and clinical implications associated with persistent neuroinflammation.

## 2. The Pathophysiology of Concussion

SRC pathophysiology has been conceptualized and has been well accepted as an issue of metabolic “supply and demand”. A biomechanical force to the head, neck, or body, resulting in axonal strain, leads to a release of glutamate, which binds to the N-Methyl D-aspartate (NMDA) receptor resulting in depolarization and ionic flux [36,37]. To regain homeostasis, adenosine triphosphate (ATP) is needed to facilitate active transport. The ionic imbalance associated with SRC results in a hyperglycolytic state which increases the need for glucose. The increased demand for the requisite substrates (e.g., glucose) to promote oxidative metabolism goes unmet as cerebral blood flow has been demonstrated to be reduced by up to 50% following a concussion in animal [22] and to a lesser extent in human models, however, the percent change, beyond statistical significance, has yet to be determined [14]. Historically, cerebral blood flow has been reported to return to normal values approximately 10 days following injury in rodent models [36,37]. However, more recent human-based studies suggest reductions in cerebral blood flow may persist beyond clinical recovery as indicated by a return to preinjury performance on clinical measures of neurocognitive function, postural stability, and symptoms used to compliment the clinical examination [14]. Stated differently, persisting deficits in cerebral blood flow have been observed in the absence of balance and neurocognitive deficits administered to high school and collegiate athletes who achieved symptom resolution at rest [14]. As the demand for glucose goes unmet, the initial hyperglycolytic state transitions to a hypoglycolytic state. The impaired metabolic state results in impaired neurotransmission, an increased vulnerability to secondary injury (e.g., recurrent concussion [38] and lower extremity injuries [39]), and impaired cognitive function (i.e., delayed reaction time and impaired memory), all of which are physiological signals of healing and are consistent with neuroinflammation [22].

To date, the body of literature surrounding clinical recovery following SRC aligns with the accepted animal-based neurometabolic cascade [24,40,41]. However, neuroimaging techniques that are based on oxygen kinetics suggest that physiological differences in humans may persist well beyond one year following injury [13,42]. The acute or long-term clinical implications associated with advanced neuroimaging findings have yet to be fully substantiated [13,43]. Our understanding of SRC-related neuroinflammation is continually evolving based on these novel physiological measures. As an alternative method to examine the physiological recovery following SRC, investigating neuroinflammation following injury using PET may help elucidate the physiological time course for recovery following SRC and assist with the identification of an objective benchmark to gauge novel therapeutic interventions.

## 3. Neuroinflammation—Microglial Responsibility

As the resident immune cells of the central nervous system (CNS), microglia play a quintessential role in the surveillance of brain parenchyma and homeostasis [44]. Approximately 5% to 10% of adult brain cells are microglia and are present in all areas of the brain; however, their population density is highly variable (whether under physiological or stress conditions) up to 10-fold in the human brain [45,46]. Anatomically speaking, increased expression of microglia cells is observed in the telencephalon and diencephalon compared to the mesencephalon or rhombencephalon, in descending order [47,48,49]. Moreover, within an anatomical region, differences in microglial populations exist between white and gray matter, as well as myelinated vs non-myelinated tissue [47,48]. Perhaps unsurprisingly, these anatomical population differences also come with functional consequences. For example, lipopolysaccharide (LPS)-induced neurotoxicity was positively correlated with microglial abundance in different regions of rat brains and, interestingly, reflected regional susceptibility differences in microglia to LPS treatment [50]. In essence, region-specific differential susceptibility of neurons to LPS is attributed to not only the population density of microglia within an anatomical region but they are also clearly responsive to concentrations of inflammation-related factors produced by these cells as well.

In general, the primary goal of the neuroinflammatory response is to mitigate external or atypical stimuli by evoking CNS immunity as a primary defense mechanism against harm and, ultimately, restore homeostasis. Unfortunately, neuroinflammation has the potential to be beneficial and damaging [51,52]. As the chief sensory mediator of brain integrity, homeostatic flux induces microglia activation through modifications in gene expression, morphology, and function [53,54,55]. Until recently, microglial activation was considered a binary phenomenon, with cells either being in a resting or activated state, which was adapted from similar nomenclature defining subpopulations of peripheral macrophages. The “activated state” of microglia was originally classified as M1 (pro-inflammatory, destructive) or M2 (anti-inflammatory, protective), with either state emerging based on exposure to certain stimuli. Each phenotype of activated microglia maintains dynamic roles in an even more dynamic course of injury resolution, involving the secretion of various neurotrophic factors and can be induced by micro- and macro-environmental stimuli. More recently, research suggests microglial activation is a dynamic and complex process [56,57]. As stressed, microglia are the chief immune modulators of the brain whose principal role is to protect the brain from injury or insult. They are the primary responders in brain inflammation, even if other cells, such as astrocytes, are also involved.

## 4. Microglia in Neurological Disorders

Microglia function to serve and protect the brain and are wholly dedicated to maintaining homeostasis. Microglia play an important role in shaping neuronal wiring in both physiologic and pathologic processes. Indeed, there is an emphasis on the role of microglia in disease pathology; however, caution is warranted to avoid distraction from the important role microglia possess in regulating and maintaining the healthy neuronal synapse [58]. For example, microglia have been shown to play a role in amyloid removal through non-saturable fluid phase micropinocytosis [59].

In response to stimuli, microglia phagocytose apoptotic neurons and secrete inflammatory factors to attract other immune cells to the site of pathology/injury. The resolution of this series of events is a tightly regulated process that occurs when an injury has been resolved [46,60]. However, if left unresolved, such as in the case of chronic or pathologic inflammation, excessive release of cytotoxic factors (reactive oxygen species, pro-inflammatory cytokines, TNF-α, etc.) results in neuronal damage and loss of brain function, which ultimately impede recovery. The persistent over-activation of microglia has been implicated in a number of CNS disorders, such as stroke, traumatic brain injury (TBI), mild traumatic brain injury (mTBI), epilepsy, Alzheimer’s disease, Parkinson’s disease, and others [46,61,62,63]. In the context of mTBI and particularly repetitive mTBI, injury-induced activation of microglia may lead to reduced dendritic branching due to disruption in homeostatic pruning, similar to that of an aging or Alzheimer’s brain [64]. Conversely, non-ramified, amoeboid-like microglia are thought to be less efficient at synaptic pruning [65], which have been observed to be highly populated in thalamic sites of injury [66]. Together, these changes in synaptic pruning resulting from mTBI may directly influence synaptic density and overall neuronal repair and health.

mTBI comprises up to 90% of all TBI incidences and is synonymous with concussion [67,68,69,70]. The temporal course of recovery from a single concussive injury, as previously mentioned, is typically short when compared to other sport-related musculoskeletal injuries. However, the cumulative effect of multiple concussions has been speculated to progress into neurodegenerative disorders [71,72]. Persistent neuroinflammation has even been observed for years after a single moderate or severe TBI [73]. Microglial activation, in addition to acute increases in cytokine production and white matter abnormalities, is a pathological hallmark of mTBI in animal models [66,74]. In rats, pro-inflammatory and anti-inflammatory microglia were shown to be upregulated (increased) following repeated mTBI (one injury every hour for four hours) within 72 h of injury; however, qRT-PCR revealed increased expression of Nos2, Ccl9, Ccl24, Cxcl2, and Ptx3 co-localized with Iba-1 staining in the cortex and hippocampus, indicating a predominance of the pro-inflammatory phenotype was observed among the total microglial population [74]. In addition, the majority of pro-inflammatory markers were resolved within 72 h following only a single concussive impact; thus, the cumulative pathologic response clearly has a longer temporal resolution course. Lafrenaye et al. also demonstrated that mTBI in adult micro pigs (a gyrencephalic animal model more akin to humans) [75,76] triggered extensive thalamic microglial activation, as determined by intense Iba-1 staining co-localized with thick, short, or absent processes. Importantly, these sites also directly correlated to the burden of thalamic axonal injury [66]. The physical relationship between activated microglia and proximal swelling of axons sustaining acute diffuse axonal injury was dramatically increased in myelinated thalamic axons compared to sham animals [66]. Collectively, acute microglial activation has been accepted as an early consequence of mTBI in animal models as well as moderate and severe TBI and may be exploited in humans during the acute phase of injury.

## 5. Neuroinflammation Biomarker—TSPO

Originally coined the peripheral benzodiazepine receptor, the translocator protein (TSPO) is an 18 kDa outer mitochondrial membrane protein. TSPO is implicated in a number of neurodegenerative functions, including redox homeostasis, modulating immune response, cholesterol transport, and steroidogenesis [77,78]. TSPO expression has been demonstrated to be significantly elevated in microglia as a result of neuronal insult. In addition, the basal expression of TSPO is relatively low in healthy brain tissue [79]; thus, TSPO has generated significant interest as an imaging biomarker for neuroinflammation. Increased TSPO expression has been correlated with immunohistochemistry validation and TSPO-specific radiopharmaceuticals using PET in a wide variety of animal models demonstrating neuroinflammation injury and disease states, including Alzheimer’s disease, stroke, TBI, mTBI, epilepsy, Parkinson’s disease, experimental autoimmune encephalitis, ALS and others [46,61,62,63,80]. These pre-clinical studies have prompted clinical research studies of numerous pathologies toward studying neuroinflammation in vivo.

The most widely studied TSPO PET radiopharmaceutical to-date is [^11^C]-*R*-PK11195, an isoquinoline derivative developed in the early 1980s [81]. While pioneering, the ultimate clinical utility of [^11^C]-*R*-PK11195 is limited by several major factors, including (1) a low brain bioavailability and poor signal-to-noise due to non-specific binding [82] and (2) the inherent short physical half-life of carbon-11 radiopharmaceuticals (*t*_1/2_ = 20 min). As such, a number of second-generation TSPO PET radiopharmaceuticals have been developed, including [^11^C]PBR-28, [^11^C]DPA-713, [^18^F]DPA-714, [^18^F]FEPPA, and others [83,84,85].

Inter-individual variability is a well-documented characteristic of second-generation TSPO ligands. These ligands have demonstrated three distinct binding patterns in human donor tissue. These three binding classifications are labeled high-affinity binders (HABs), low-affinity binders (LABs), or mixed affinity binders (MABs), where MABs represent a TSPO receptor expressing one class of sites with an affinity approximately equal to the mean of those for HABs and LABs [86]. Second-generation TSPO ligands have demonstrated a distribution pattern of approximately 50% to 65% HABs, ~30% MABs, and 5% to 25% LABs [86,87,88]. The binding affinity of second-generation TSPO radiopharmaceuticals can be predicted by a single nucleotide polymorphism (SNP) from alanine to threonine at position 147 in TSPO’s fifth membrane domain [89]. This polymorphism, allele A, alters mitochondrial TSPO protein structure from the ancestral G allele. Humans can be grouped as HABs (GG), MABs (AG), or LABs (AA). As such, patients undergoing a TSPO imaging study should first be screened for the identification of their TSPO genotype. Stratification of patients by genotype or excluding LABs altogether has been demonstrated to yield more consistent quantitative comparisons with second-generation TSPO radiopharmaceuticals [90,91,92]. A recent study demonstrated an allosteric interaction with [^3^H]DPA-713 in the presence of cholesterol, or PK11195, specifically observed in LAB-TSPO platelets, which was not observed in MAB or HAB-TSPO platelets. These data provide a molecular basis for LAB-TSPO ligand interactions and further support the exclusion of subjects bearing the LAB genotype from TSPO PET imaging studies with second-generation ligands [93].

Importantly, TSPO expression has been characterized in astrocytes and endothelial cells and cannot differentiate microglia from infiltrating macrophages. As such, TSPO PET may be more appropriate to quantify microglial density rather than the activation state of microglia [94,95,96]. Furthermore, the precise functional relevance of TSPO, particularly in the context of SRC-associated neuroinflammation, is not fully understood and warrants further study. To assess biological signatures specifically associated with microglial activation, many other biomarkers have been studied to-date, including P2X_7_R [97], P2Y_12_R [98], COX1/2 [99,100], and CSF-1R [101]. However, further clinical research studies are needed to validate these biomarkers in the context of disease.

## 6. Imaging Neuroinflammation—(m)TBI

A major impediment to advancing mTBI imaging research is the inherent complexity of SRC itself, as well as the temporal course of the injury within a given individual or cohort of subjects. The primary injury is an initial biomechanical insult that involves kinetic energy-driven neuronal, axonal, and vascular damage [21]. This initial insult induces a cascade of secondary sequelae leading to excitotoxicity, oxidative stress, apoptosis, autophagy, necrosis, and more. Thus, TBI has acute and chronic disease characteristics implicating pathophysiological and neuroinflammatory consequences in the brain. Further complicating the clinical interpretation of neuroinflammation, as observed using advanced neuroimaging techniques, is a temporal dichotomy of beneficial and destructive milieu. The latter may further exaggerate the primary injury [80,102,103,104].

Axonal abnormalities and neuronal degeneration have been associated with activated microglial regions in the brain [66,105,106]. Most often, studies in animal models and humans have detected microglial activation early after TBI [80,103]; however, microglial activation has been noted to persist for years beyond the initial injury [73]. [^11^C]-*R*-PK11195 was studied in patients following moderate to severe TBI from a range of 11 months to 17 years post-injury and found binding to be significantly increased in the thalamus, occipital cortices, putamen, and posterior limb of the internal capsules, but noted microglial activation is not persistently increased around the site of focal brain lesions [107]. In addition, increased thalamic binding correlated with a degree of cognitive impairment. These observations are consistent with other findings noting thalamic damage is a common occurrence in human TBI [66,108,109]. The prominence of microglial activation in subcortical structures following TBI has been suggested to reflect their dense connectivity and implies microglia behave differently locally at the site of injury from those at remotely connected structures. Thus, this may reflect a slowly progressive process within damaged white matter similar to what is observed in stroke [110,111].

A recent study investigating [^11^C]DPA-713 binding in former NFL football players whose self-reported number of concussions ranged from 0 to 40 and the span of years since retirement from play ranged from 24 to 42 years. The study revealed significantly higher total distribution volume (Vt) in the hippocampus, amygdala, supramarginal gyrus, and temporal pole compared with healthy age-matched controls [73]. A follow-up study by Coughlin et al. examined active (*n* = 4) and former (*n* = 10) NFL football players with a range of 1 to 21 years since their last self-reported concussion. The data revealed elevated/increased binding (24–41% increase) of [^11^C]DPA-713 in the bilateral hippocampus, left entorhinal cortex, bilateral parahippocampal cortices, and bilateral supramarginal gyri compared to healthy age-matched controls [27]. Increased binding was also independent of age group (active vs. retired play), ethnicity, body mass index, and years of education. Taken together, these studies demonstrate increased neuroinflammation in NFL players across a wide range of ages and participation. These findings are important in demonstrating cerebrovascular disease is not a dependent variable on TSPO binding as might be observed in aged subjects with regional brain atrophy. However, the authors acknowledge the mean reported time-lapse from self-reported concussion to be seven years. Thus, the temporal relationship between mTBI and TSPO in impact sports remains to be elucidated.

We recently studied microglial activation in National Collegiate Athletics Association (NCAA) athletes who had sustained a sports-related concussion (SRC) [112]. The study enrolled both male and female subjects at the time of being clinically diagnosed with an SRC, then upon being granted “return-to-sport” status from the gold-standard clinical battery, subjects participated in a 90 min dynamic PET/CT scan using the TSPO imaging agent, [^18^F]DPA-714. All subjects were matched by TSPO genotype and non-athletes of similar age. Using a robust advanced imaging analysis pipeline (Figure 1), we calculated the Vt of 164 brain regions among all participants. Despite achieving clinical recovery, athletes demonstrated 70–115% increases in [^18^F]DPA-714 Vt relative to age-matched controls, regardless of sex, most consistently in the limbic system, dorsal striatum, and medial temporal lobe. Static PET images from 45–60 min post-injection of [^18^F]DPA-714 also revealed striking differences between concussed athletes and age-matched controls (Figure 2). Importantly, no notable differences were observed between clinical measures of SRC and Vt between athletes and age-matched controls at the time of PET imaging. This discordance spurs several questions regarding the resolution of microglial activation relative to the resolution of clinical symptoms, such as whether the observed increase in PET signal is associated with a natural pro- and/or anti-inflammatory response of repair or the initiation of a chronic degenerative neuroinflammatory cascade. Further studies are warranted to understand the temporal resolution of microglial activation following SRC, the impact on microglial activation after sustaining multiple SRCs, and whether returning to play before the resolution of microglial activation increases susceptibility to further injury.

An additional study used [^123^I] Single Photon Emission Computed Tomography (SPECT) to examine I-CLINDE binding in patients presenting to the emergency department after mTBI [29]. In this study, patients were imaged at both 1 to 2 weeks and 3 to 4 months post-injury. Comparing 14 mTBI subjects against 22 healthy controls, Ebert et al. found all subjects who experienced mTBI had significantly increased V_t_ at 3 to 4 months in the seven patients presenting with post-concussion symptoms (PCS), while no signs of structural damage, including diffuse axonal injury, were noted on MRI. Interestingly, no significant differences in V_t_ were observed between mTBI patients considered to have “good recovery” and patients with PCS. Further studies are needed to understand the acute temporal course of TSPO following a single mTBI, if peripheral inflammatory factors associate with microglia activation, the temporal TSPO status of athletes in contact sports independent of mTBI, and the relationship of TSPO in athletes who experience mTBI from high-contact and low-contact sports. In addition, the ability to segment neuroprotective from detrimental neuroinflammation in vivo would unquestionably impact the management of mTBI.

Several other biomarkers have been investigated for in vivo imaging to increase our understanding of the underlying mechanisms of the spectrum of TBI. One of the most commonly studied radiopharmaceuticals is 2-[^18^F]fluoro-2-deoxy-glucose ([^18^F]FDG or FDG). Not only is FDG widely available, facilitating clinical research studies, but FDG-PET can also be used, much like fMRI, to study subjects in a resting state or when responsive to a stimulus [113,114,115]. Impaired glucose metabolism has been observed in veterans sustaining blast injuries, notably in the mesencephalon and rhombencephalon [116]. The authors noted nearly all veterans enrolled in the research study (10/12) also met diagnostic criteria for PTSD, potentially confounding psychiatric diagnosis, and blast exposure effects on glucose metabolism. However, other studies [117,118] have noted glucose metabolism to be independent of PTSD in combat war veterans, suggesting PTSD comorbidity is not a confounding effect. Follow-up studies in combat veterans continue to explore whether persisting cognitive deficits and post-concussive symptoms from mTBI are associated with enduring structural/functional brain abnormalities versus the comorbidity of depression or PTSD [119]. Accordingly, FDG-PET has linked mTBI glucose metabolism alterations in brain regions to similar profiles observed in more severe cases of TBI, and some of these studies suggest hypometabolism in specific areas (frontal and temporal regions) that correlate with deficits in neurobehavioral function and neuropsychological testing [115,120]. However, the use of FDG-PET in studying CNS pathology is ultimately limited by low brain contrast and warrants study with PET agents with higher specificity, such as TSPO, for biological markers of disease.

## 7. Analytical Concerns—Dynamic vs Static Imaging

In static PET (sPET), the scanner maps the volumetric concentration of radioactivity within a short sampling window, typically 45 min after injection (Figure 3A). The key variable returned by sPET is the standardized uptake value (SUV). SUV, measured at a specific time point after tracer injection, provides a semi-quantitative snapshot of tracer activity [121]. Dynamic PET (dPET), on the other hand, measures radioactivity on a continuous basis (Figure 3B) [122]. In this paradigm, the subject is scanned at the pre-injection baseline, and the tracer is injected during active scanning. Scanning continues akin to a series of movie frames that capture the changing concentrations of radiotracer per scanning interval. The process provides a volumetric concentration-time profile for tracer uptake and metabolism; kinetic models measure, voxel-by-voxel, the rate, and distribution of radioactivity uptake and decay.

PET imaging has become an increasingly useful research modality in neuroimaging as it can visualize molecular processes, such as neuroinflammation (and microglial activation), in vivo, longitudinally, and noninvasively [123,124]. Reversible and non-reversible 2-tissue (2T) kinetic models have been used in several pre-clinical and clinical indications related to fluorine-18 labeled radiopharmaceuticals over the last decade to analyze time-resolved (dynamic) PET data. Dynamic [^18^F]FDG (FDG) PET utilizes a non-reversible 2T model, as FDG is phosphorylated by hexokinase and gets trapped as FDG-6-phosphate within the cell to compute cerebral glucose metabolism. Recent work from our laboratory computed cerebral FDG uptake rates in a mouse model of dystonia utilizing the 2T non-reversible model [125]. However, the application of reversible 2T compartment models is considered optimal in the quantification of cerebral [^18^F]DPA714 binding in healthy controls [126], in patients with Alzheimer’s Disease [127,128] and multiple sclerosis [129], as this tracer measures receptor binding, which is a reversible process.

An important component in the compartment models (both reversible and non-reversible) is the blood input function, which is defined as the amount of tracer in the blood that is available for the tissue to use [130,131]. Recent studies using [^18^F]DPA-714 and [^11^C]DPA713 [28,123] derived the blood input function from arterial blood sampling over the whole period of PET acquisition (60–90 min). This could be challenging in both pre-clinical and clinical settings as arterial blood sampling involves catheterization of the carotid or femoral artery, is highly invasive, and adds additional complexity to the research study. The net result is an unreliable computation of kinetic rate constants and hence net tracer influx constants or total distribution volumes. Image-derived blood input function (IDIF) has been developed by several laboratories [130], including ours, wherein we use the imaged inferior vena cava [132] or the left ventricular blood pool [133] as sources of IDIF for computing cerebral FDG uptake rates in rodents [125]. In PET scanners [16], the limited field of view (~23 cm) allows for imaging only the brain. As a result, only the carotid arteries in the brain (external or internal) can be used as sources for IDIF. In addition, PET signal contributions from blood metabolites may contaminate the blood, further confounding the computation of the IDIF [28]. Recent work from our laboratory using [^18^F]DPA714 in SRC athletes utilized late time point venous samples to correct IDIF from the carotid arteries for computation of total tracer distribution volumes [112] in a graphical Logan model [134]. Reference tissue models [135] obviate the need for image-derived blood sampling from the carotid arteries, wherein the cerebellar grey matter serves as a reference region and has been found to reliably assess binding potential in Alzheimer’s disease [127]. Further research is warranted to determine a suitable reference region for TSPO-targeted radiopharmaceuticals, which may further simplify and deploy this method for SRC and other neurological/neurodegenerative disorders with neuroinflammation.

## 8. Magnetic Resonance Imaging

Over the past two decades, functional magnetic resonance imaging (fMRI) has been commonly used to study in vivo changes in human brain “activity” following mTBI, as discussed in prior reviews [17,136,137,138,139]. fMRI studies of mTBI have been conducted in different age groups (pediatric to adult), at different time lags after injury (days to years), and in different injury settings (e.g., sports, emergency room, or military). Study designs include task-related fMRI examining statistical correlations between MRI signal and specific cognitive processes (most frequently working memory) and resting state fMRI examining temporal synchrony of MRI across different anatomic regions to identify patterns of co-activation. Collectively, these fMRI studies show broad differences in fMRI activation between mTBI and healthy control (HC) groups, but the nature of the activation (hypoactivation vs hyperactivation) and anatomic locations of differences described vary broadly across studies. The differences in the location of activation may be an emergent property of spatial heterogeneity of mTBI in individual patients [140]. But any account of discrepancy in the nature of activation must consider that fMRI measures blood oxygen level-dependent (BOLD) signal as a proxy for underlying neuronal activity; stated simply, BOLD fMRI does not directly measure neuronal activity.

BOLD fMRI relies on a neurovascular coupling mechanism where changes in local neuronal activity are linked to local decreases in paramagnetic deoxyhemoglobin (HbR), a phenomenon measurable as T2* relaxation in the setting of fMRI [141]. It has been well described that local neuronal activity initiates a metabolic cascade of events, resulting in increased local blood flow, increased local oxyhemoglobin (HbO), and decreased local HbR, peaking at 3–5 s after onset of neuronal activity and then slowly returning to baseline [142]. In the BOLD fMRI literature, this characteristic response is called the “hemodynamic response function” (HRF). At a cellular level, the neurovascular coupling cascade is likely mediated by a collection of astrocytes, pericytes, and interneurons, interacting in concert with vascular endothelial cells via a variety of biologic pathways [143]. The classical interpretation of BOLD fMRI depends on the assumption that neuronal activity causes a linear time-invariant, spatially-invariant change in the BOLD signal [144]. This specific assumption has been strictly proven false but close enough to true that it supports a large body of cognitive neuroscience research in healthy control participants; how the assumption fares in situations of brain pathology are relatively underexplored. Thus, BOLD fMRI findings in mTBI are collectively limited by their ability to ascribe changes to a specific physiologic mechanism. In short, a conclusive and comprehensive demonstration of the effects of neuroinflammation on the neurovascular coupling mechanism underlying BOLD fMRI signal remains a notable gap in the literature.

Diffusion MRI (dMRI) quantifies the directional movement of water molecules as constrained by cellular membranes. dMRI is particularly well suited to study neuronal axons as the impermeable myelin sheaths restrict molecules from dispersing radially, so they instead move along the length of the axon. This allows dMRI to be more sensitive to subtle cases of axonal damage or neuroinflammation than traditional structural MRI or CT, including in mTBI [145,146,147]. The complex structures and properties of axons leave them particularly vulnerable to mechanical damage from TBI [18,148], and diffusion tensor-derived measures of axonal health, such as fractional anisotropy (FA) and the closely related measure of mean diffusivity (MD), have consistently shown differences between controls and individuals with mTBI [149,150,151]. As more advanced dMRI analysis techniques continue to be developed, it is important for clinicians and researchers to be knowledgeable about the most common techniques and their application in mTBI patients. Changes in dMRI measures can have multiple underlying causes [152,153], and it is important to consider multiple approaches and contextual factors in order to compose a complete picture of brain changes due to mTBI.

Diffusion tensor imaging (DTI) is the most widespread model used to analyze diffusion data and essentially involves three directional vectors, the magnitude of each corresponding to how far water molecules diffuse unimpeded in that direction. Results from DTI studies on mTBI have been mixed, with differing results among patients after single or multiple impacts and if a patient is in the acute phase of injury or is in the late stage of recovery. For example, a review of acute and semi-acute mTBI (defined by the authors as less than three months since injury) by Dodd et al. [154] found that 14 studies reported decreased fractional anisotropy (FA); 16 reported increased FA; 2 reported ROI dependent bidirectional changes in FA [155,156]; and 5 reported no significant changes in FA [157,158,159,160,161]. The range of results presented is perhaps most reflective of the difficulty in studying a highly complex and variable injury, with time-dependent symptoms, in human patients with heterogeneous injuries and lack of specificity. The importance of rigorous design and control is reinforced by a systemic review of adolescent mTBI by Schmidt et al. [149]. The authors reported nine DTI studies, all of which detected significant differences between mTBI and control groups. However, a limitation of these studies was a limited sample size and methodological considerations which reinforces the need for further research.

Nevertheless, there is a considerable body of evidence that DTI can detect alterations in a consistent series of regions following mTBI. The corpus callosum [146,150,156,160,162,163,164,165,166,167,168] and thalamus [160,168,169,170,171] are among the most frequently significant WM and subcortical ROIs, depending on the parcellation method. A systematic review by Khong et al. has also established that DTI abnormalities correlate with both incidence and severity of post-concussion syndrome, though there was no consensus as to the DTI measure or ROI that drove the association, further emphasizing the heterogeneous nature of mTBI [150]. The relationship of DTI measures to neuroinflammatory processes is also complicated. In a PET/DTI study looking at 10 individuals 11 to 204 months post-TBI, there was significantly increased MD and significantly decreased FA in the corpus callosum compared to age-matched controls; however, these DTI changes did not correlate with CNS microglial activation, as measured by PET [107]. Acute neuroinflammation can induce a complex cascade of cellular changes, which may also be detectable by DTI, but interpreting these findings is frequently hampered by a lack of validated specific information about which individual cellular processes are occurring or co-occurring [172].

## 9. Clinical Implications

During the past decade, investigations of neuroinflammation using central and systemic biomarkers have advanced our understanding of SRC beyond the neurometabolic cascade. Despite our increased understanding of the secondary neuroinflammatory response, this exciting body of research is still in its infancy. For example, when discussing the promise of IL-6 as a biofluid marker of SRC, Meier et al. conclude that neuroinflammatory biofluid markers may hold promise in the diagnosis of SRC when used in concert with standard-of-care clinical measures of SRC [19]. However, circulating peripheral markers may or may not correlate with a specific microenvironmental status within the CNS; thus, additional research is needed to correlate these findings. To this end, an increasing body of evidence is supportive of the roles of advanced neuroimaging techniques such as PET, fMRI, and DTI in tracking physiological recovery from SRC. However, disparate findings suggest that future studies with adequately powered samples, the replication of methodologies, an extension of established methodologies to include female participants, and more thorough medical histories are critical in understanding the role of neuroinflammation following SRC. The simplest questions and potentially the most complex to answer as it relates to neuroinflammation following SRC are what a normal neuroinflammatory response is and which phenotypic changes are clinically meaningful.

## 10. Conclusions

To date, the collective body of evidence as reviewed in this manuscript supports the presence of neuroinflammation acutely following a diagnosed SRC [19,31,32], throughout recovery [32], and potentially many years beyond [13,27,28]. These findings are apparent even after complete clinical recovery and successful RTS [173]. These signs of persistent neuroinflammatory response to SRC may be associated with the suggested increased risk of musculoskeletal injury [174], mild cognitive impairment [2], and other speculated mid- to late-life consequences [175,176]. It is also recognized that inflammation is an essential mechanism in human health and disease. Thus, the mere presence of inflammation following SRC may be expected as a natural progression of injury resolution. Further research is needed to better coalesce clinical symptom recovery and physiological recovery; in addition, the ability to delineate the expected pro-inflammatory response from a more destructive, chronic condition would have an enormous impact on SRC management. Important follow-up studies, such as longitudinal evaluation of changes in TSPO PET signal after a subject experiences a single SRC are warranted. In addition, appropriately powered studies quantifying normal TSPO PET signal in athletes/individuals engaging in activities predisposed to SRC/mTBI relative to non-sport/normal healthy control subjects will be a critical step toward understanding physiological recovery as it relates to TSPO PET. As previously mentioned, this body of research, as well as the methodology and technology used to investigate neuroinflammation following SRC is in its infancy. However, findings of this collective body of work will eventually help us understand the physiological recovery of SRC, the clinical measures that associate with it, and assist the identification of targeted pharmaceutical interventions that may ameliorate the negative consequences that have been associated with this injury.

## Data Availability

Data used for figures in this review article were generated by the co-authors and are available upon request.

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
