# Peer review of "Concussion: Beyond the Cascade"

_cells, 2023, doi:10.3390/cells12172128_

Round 1

Reviewer 1 Report

Overview: Very important topic of research that is affecting the lives of athletes in the short and long term and is a cause of many neurological sequelae in later life. The authors make some poignant points about the role of inflammation in mTBI, considerations for the investigation and analysis of inflammation using imaging, and the lack of concordance between systemic and central inflammation. Overall, this well written review on an important are of research. 

Please see below for minor comments for each section. 

1.     Introduction.

The introduction nicely outlines the acute and potential chronic effects of SRC, and clinical approaches for diagnosis and monitoring. A nice graphic was used to illustrate the separation of clinical vs physiological recovery, however further information on the specifics of these terms in the context of SRC is warranted. It is unclear what the definition of these terms represent – e.g., clinical recovery would suggest full clinical symptom reversal but this is in contrast to the persistent clinical symptoms outlined in the first paragraph. Additional information on the assessment tools mentioned and advanced imaging methods would also provide more context for general readers who may not be familiar with these terms.

Sport Concussion Assessment Tool (SCAT3), the Standardized Assessment of Concussion (SAC), the Balance Error Scoring System (BESS), and the Automated Neuropsychological Assessment Metrics (ANAM) computerized neurocognitive assessment.” 

advanced neuroimaging (global functional connectivity, arterial spin labeling, fractional anisotropy, and mean diffusivity).

Similarly, the “metabolic crisis” and its link to neuroinflammation was mentioned but it was not explained. A sentence or two doing so would be beneficial.

In paragraph 4 where biofluid markers of inflammation are discussed, please consider a short summary of the main biomarkers investigated in the context of mTBI.

2.     The Pathophysiology of Concussion

Section 2 would benefit from providing the % change o fblood floe in hums from reference 13. 

“…cerebral blood flow has been demonstrated to be reduced by up to 50% following a concussion in animal[14] and to a lesser extent in human models.[13]”

Similar comment to the introduction where more clarity on the definition of clinical recovery would benefit the text. 

Example: from the below sentence it is not clear what defines clinical recovery if clinical symptoms are persisting. 

“However, more recent human-based studies suggest reductions in cerebral blood flow may persist beyond clinical recovery as indicated by clinical measures of symptoms, neurocognitive function, and postural stability.” 

3.     Neuroinflammation – Microglial Responsibility 

Nice brief summary of microglial function in homeostatic and inflammatory conditions. The authors highlight the complexity of microglial activation. 

Extra spacing on line 162 and in a few other places throughout the manuscript.

4.     Microglia in Neurological Disorders

Considering the authors highlighted the complexity of microglial activation in the previous section. This section would greatly benefit from additional details on what the pro- and anti-inflammatory markers that were used to define this activation states were in the following text. 

“In rats, pro-inflammatory and anti-inflammatory microglia were shown to be upregulated (increased) following repeated mTBI (one injury every hour for four hours) within 72 hours of injury; however, a predominance of the pro-inflammatory phenotype was observed among the total microglial population.[59] In addition, the majority of pro-inflammatory markers were resolved within 72 hours following only a single concussive impact, thus, the cumulative

pathologic response clearly has a longer temporal resolution course.”

Similarly, what markers were being used to define microglial activation in the following sentences.

Lafrenaye et al also demonstrated that mTBI in adult micro pigs (a gyrencephalic animal model more akin tohumans)[60, 61] triggered extensive microglial activation in thalamic sites, which also directly correlated to the burden of thalamic axonal injury.[58]”

5.     TSPO

Nice summary of TSPO and the functions that are associated with it. However, it is worth noting that the exact function of TSPO remains unknown and in particular relevance to this review, the function within the context of neuroinflammation is not understood. 

More recent publications indicate that TSPO reflects microglial density not activation state which is worth noting in this section. 

https://www.ncbi.nlm.nih.gov/pmc/articles/PMC8453709/

https://academic.oup.com/brain/article/142/11/3440/5580333

https://www.ncbi.nlm.nih.gov/pmc/articles/PMC6308309/

Additionally, when describing TSPO as a biomarker it is important to note the issue of cellular and functionally specificity – it is also expressed on astrocytes and endothelial cells, and cannot differentiate microglia from macrophages. In essence, the limitations and benefits of TSPO as an imaging biomarker warrants further discussion here. 

Authors should also consider mentioning that TSPO is one of many biomarkers of neuroinflammation being investigated worldwide, albeit still the most widely investigates to date. 

The authors state that “the basal expression of TSPO is relatively low,[64] thus, TSPO has generated significant interest as an imaging biomarker for neuroinflammation” however, this low TSPO expression is only pertinent to healthy brain tissue. TSPO basal expression in other organs (e.g., kidneys, lungs, etc.) is moderate/high.  

Missing bracket for [18F]DPA-714 on line 214

6.     Imaging Neuroinflammation – (m)TBI

Nice explanation of FDG-PET. It would be worth mentioning the use of FDG-PET to investigate inflammation by some groups and the idea of glucose metabolism reflecting increased microglial activity. 

To enhance flow and continuity of  the topic of TSPO-PET, the authors should consider reordering the paragraphs in sections 5 and 6 so that the TSPO-PET imaging follows the TSPO biomarker section or providing some context for readers as to why there is a TSPO section first – i.e., explain how TSPO-PET is commonly used for imaging neuroinflammation in mTBI.

When describing the TSPO-PET studies in NFL players please provide the relative increases (number or percentages as reported in the NCAA studies. Please also add “elevated/increased” to the following sentence: 

“The data revealed significant binding of [11C]DPA-713 in the bilateral hippocampus, left…”

Consider rephrasing the following sentence as both pro- and anti-inflammatory responses can occur during acute and chronic neuroinflammatory conditions. 

“such as whether the observed increase in PET signal is associated with a pro-inflammatory (acute repair) or anti-inflammatory (chronic destructive) response.”

7.     MRI

Really nice explanations of fMRI, dMRI, and DTI. 

On line 480, the author might mean systematic review instead of “systemic review”.

Author Response

Thank you for your thorough review and constructive comments to increase the quality of our manuscript. We have addressed all critiques, point-by-point, in the attached file.

Reviewer 2 Report

The current review by Neumann et al. centers on understanding the cellular processes associated with concussion and focuses on imaging studies aimed to determining damage elicited by repeated head injury and concussions. The manuscript is very well written, and the authors should be commended for the timely production of this review. There are a few minor issues however that should be addressed prior to publication.

1.    A little more detail regarding the activation states of microglia may be a good addition to the manuscript. A large part of the imaging section and review focuses on inflammation and there is but a small mention of the activation states that comprise microglia within the CNS. This is a varied response and there should be more detail outlining this process.

2.    Along the same lines, there is a quick mention of microglia phagocytosing apoptotic neurons, however this field is very complex with microglia also involved in synaptic pruning and synaptic regulation. This section from around line 150 to line 165 or so should be elaborated on to relay to the reader the full complexity of this process and how it may tie into concussion/mTBI.

3.    Line 433: The sentence should read, “cells via a variety of biologic pathways,” instead of “pharmacologic pathways,” as the sentence is not necessarily referring to the exogenous manipulation of these pathways with drugs or drug candidates.

4.    Lines 512 through around 525 discuss inflammation and other health issues. It begs the question as to what the levels of inflammation are that are perhaps “normal” or within a defined range and how this will apply to the future use of imaging and other techniques in the assessments of inflammation and neuroinflammation. Many of the data currently available are preclinical in this area and many things are noted to affect aspects of neuroinflammation, including aging, disease, other forms of trauma etc. Is there anything to be added discussing levels of “normal” inflammation in the populations mostly at risk for mTBI/concussion?

Author Response

(The authors gave the same response as above.)

Reviewer 3 Report

No comments or suggestions.

Author Response

Thank you for your time and thorough review.